

# Testing the short-and long-term effects of elevated prenatal exposure to different forms of thyroid hormones

Tom Sarraude[1,2], Bin-Yan Hsu[1], Ton Groothuis[2] and Suvi Ruuskanen[1]

[1] Department of Biology, University of Turku, Turku, Finland
[2] Groningen Institute for Evolutionary Life Sciences, University of Groningen, Groningen, Netherlands

Corresponding author
Tom Sarraude, t.sarraude@rug.nl

## ABSTRACT

Maternal thyroid hormones (THs) are known to be crucial in embryonic development in humans, but their influence on other, especially wild, animals remains poorly understood. So far, the studies that experimentally investigated the consequences of maternal THs focused on short-term effects, while early organisational effects with long-term consequences, as shown for other prenatal hormones, could also be expected. In this study, we aimed at investigating both the short- and long-term effects of prenatal THs in a bird species, the Japanese quail *Coturnix japonica*. We experimentally elevated yolk TH content (the prohormone $T_4$, and its active metabolite $T_3$, as well as a combination of both hormones). We analysed hatching success, embryonic development, offspring growth and oxidative stress as well as their potential organisational effects on reproduction, moult and oxidative stress in adulthood. We found that eggs injected with $T_4$ had a higher hatching success compared with control eggs, suggesting conversion of $T_4$ into $T_3$ by the embryo. We detected no evidence for other short-term or long-term effects of yolk THs. These results suggest that yolk THs are important in the embryonic stage of precocial birds, but other short- and long-term consequences remain unclear. Research on maternal THs will greatly benefit from studies investigating how embryos use and respond to this maternal signalling. Long-term studies on prenatal THs in other taxa in the wild are needed for a better understanding of this hormone-mediated maternal pathway.

## INTRODUCTION

Maternal effects represent all the non-genetic influences of a mother on her offspring and have received increasing attention in evolutionary and behavioural ecology. Through maternal effects, mothers can influence the fitness of their progeny by adapting their phenotype to expected environmental conditions ("adaptive maternal effects" in *Marshall & Uller (2007)* and *Mousseau & Fox (1998)*), and this view is now also incorporated in the human disease literature (*Gluckman, Hanson & Spencer, 2005*). Maternal hormones transferred to the offspring can mediate important maternal effects. Historically, research

on maternal hormones has mostly focused on steroid hormones (*Groothuis et al., 2005*; *Von Engelhardt & Groothuis, 2011*). While research on maternal thyroid hormones (THs) has emerged between the 80s and the 90s in several taxa (mammals, *De Escobar et al., 1985*; fish, *Brown et al., 1988*; birds, *Wilson & McNabb, 1997*), these hormones are still underrepresented in the literature on hormone-mediated maternal effects (reviewed in *Ruuskanen & Hsu (2018)*).

Thyroid hormones are metabolic hormones produced by the thyroid gland and are present in two main forms: the prohormone thyroxine ($T_4$) and the biologically active form triiodothyronine ($T_3$). THs play a crucial role in various aspects of an individual's life, for example, development, metabolism and reproduction, across vertebrates, including humans (*Morreale de Escobar, Obregon & Escobar del Rey, 2004*; *Krassas, Poppe & Glinoer, 2010*). In humans, physiological variation of maternal THs (i.e., no clinical symptoms in both mothers and foetuses) is found to be associated with infant birth weight and IQ in older children (*Medici et al., 2013*; *Korevaar et al., 2016*). In other vertebrates, THs in general play a role in brain development and neuronal turnover (mammals, *Morreale de Escobar, Obregon & Escobar del Rey, 2004*; birds, *McNabb, 2007*). THs control the endothermic heat production, and are therefore important in thermoregulation in homeothermic species (mammals, *Danforth & Burger, 1984*; birds, *McNabb & Darras, 2015*).

THs can act, in concert with other hormonal axes, as mediators of life stage transitions across vertebrates (reviewed in *Watanabe, Grommen & De Groef (2016)*). The interaction between THs and corticosteroids on amphibian metamorphosis is a well-known example of such effect on life stage transition (*Kikuyama et al., 1993*; *Wada, 2008*). THs are involved in gonadal development, and hyperthyroidism tends to accelerate maturation (*Holsberger & Cooke, 2005*), and coordinate the transition between reproduction and moult (*McNabb & Darras, 2015*). Administration of exogenous THs is known to stop egg laying and induce moult in birds (*Sekimoto et al., 1987*; *Keshavarz & Quimby, 2002*). THs are also involved in the photoperiodic control of seasonal breeding (*Dardente, Hazlerigg & Ebling, 2014*). For example, thyroidectomised starlings transferred to long photoperiods became insensitive to future changes in photoperiod, and short photoperiod did not induce gonadal regression (*Dawson, 1993*).

While there has been recent research effort on the influence of maternal THs on offspring traits across vertebrate taxa, there are still substantial gaps in our knowledge. Manipulating yolk hormones within the natural range of a species is necessary to better understand the role of maternal THs in an eco-evolutionary context. In humans, studies have essentially looked at the consequences of clinical hyper- or hypothyroidism (but see *Medici et al., 2013*). Research in fish has applied supra-physiological doses for aquaculture purposes (*Brown et al., 2014*). However, these studies do not give information on how variations within the natural range of the species would shape offspring phenotype and affect its fitness, in turn influencing evolution. While recent literature on birds has shown that even physiological variations of prenatal THs can have phenotypic consequences (*Ruuskanen et al., 2016*; *Hsu et al., 2017*, *2019*; *Sarraude et al., 2020*), this view is still underrepresented in maternal THs research.

Besides, research on maternal THs up to date has mainly investigated the short-term effects of prenatal THs on developing fish (*Brown et al., 1988*; *Raine et al., 2004*) and amphibians (*Duarte-Guterman et al., 2010*; *Fini et al., 2012*) and pre-fledging birds (*Ruuskanen et al., 2016*; *Hsu et al., 2017*, *2019*; *Sarraude et al., 2020*). So far, only a study on rock pigeons has looked at the influence of yolk THs on post-fledging survival and found no effect (*Hsu et al., 2017*). None of these studies in any taxa investigated the potential organisational effects of prenatal THs on life-history stage transitions in adult life. Early exposure to elevated THs may affect the hypothalamic-pituitary-thyroid (HPT) axis (humans and mice: *Alonso et al., 2007*; *Srichomkwun et al., 2017*; *Anselmo et al., 2019*), via epigenetic modifications for example, such as those induced by adverse early life conditions (*Jimeno et al., 2019*) or yolk testosterone (*Bentz, Becker & Navara, 2016*).

Oviparous species, such as birds, are suitable models for studying the role of maternal hormones on the progeny because embryos develop in eggs outside mothers' body. The content of an egg cannot be adjusted by the mother after laying, which facilitates the quantification of hormones transmitted by the mothers. In addition, the measurement and experimental manipulation of maternal hormones in the egg after it has been laid is not confounded by maternal physiology. These advantages combined with their well-known ecology and evolution, birds have become the most extensively studied taxa in research on the function of maternal hormones (*Groothuis et al., 2019*).

Previous studies on prenatal THs in birds focused only on altricial species (great tits, *Ruuskanen et al., 2016*; rock pigeons, *Hsu et al., 2017*; collared flycatchers, *Hsu et al., 2019*, pied flycatchers, *Sarraude et al., 2020*). Embryonic development differs substantially between altricial and precocial species. In the latter, embryonic development is more advanced than in the former. In addition, precocial embryos start their endogenous production of TH around mid-incubation, considerably earlier than their altricial counterparts, in which endogenous TH production begins only after hatching (*McNabb, Scanes & Zeman, 1998*). While embryonic hormone production may limit the influence of maternal hormones, prenatal hormones have been shown to affect chick endogenous production and sensitivity (*Pfannkuche et al., 2011*). Overall, exposure to maternal hormones may be of different importance in these two developmental modes.

Previous research has studied the effects of $T_3$ only (*Raine et al., 2004*; *Walpita et al., 2007*; *Fini et al., 2012*) or a combination of $T_3$ and $T_4$ (*Ruuskanen et al., 2016*; *Hsu et al., 2017*, *2019*; *Sarraude et al., 2020*), where the effects of the two forms cannot be separated. Although $T_3$ is the biologically active form that binds to the receptors, both $T_3$ and $T_4$ are deposited in eggs (*Prati et al., 1992*) and $T_4$ may be converted to $T_3$ via deiodinases from the mother or the developing embryo (*Van Herck et al., 2015*) or may still exert non-genomic actions (reviewed in *Davis, Goglia & Leonard (2016)*). Manipulating yolk $T_4$ and $T_3$ independently would help understanding the relative contribution of these two hormones.

In this study, we aimed at assessing the effects of maternal THs on development and life-history traits in a precocial bird species, the Japanese quail (*Coturnix japonica*). We manipulated eggs with either an injection of $T_4$ or $T_3$ separately, a combination of both hormones, or a control injection of the vehicle saline solution. First, we hypothesise that
elevation of yolk THs in Japanese quails positively affects hatching success, as found in two studies on collared flycatchers and rock pigeons (*Hsu et al., 2017*, *2019*, but see *Ruuskanen et al., 2016* and *Sarraude et al., 2020*). Second, elevation of yolk THs is predicted to increase the proportion of well-developed embryos before hatching, as found in rock pigeons (*Hsu et al., 2017*). We therefore looked at the age at mortality in unhatched eggs. Third, we expect elevated yolk THs to affect chick growth (in body mass, tarsus and wing length) either positively (*Wilson & McNabb, 1997*; *Hsu et al., 2019*; weak effect in *Sarraude et al. (2020)*), negatively (*Hsu et al., 2017*), or in a sex-specific manner (*Ruuskanen et al., 2016*). Prenatal THs may exert most of their effects in the offspring early life; this is why we separately tested both posthatch morphological traits and the growth curve. Similarly, we also independently analysed morphological traits at adulthood, as these traits may affect the fitness of an individual. For example, small adult females may lay smaller eggs and larger males may be more dominant. Fourth, we predict that yolk THs will have organisational effects on life-history stage transitions; that is, age at sexual maturity and male gonadal regression (using cloacal gland size as a proxy), and moult when birds are exposed to short photoperiod. Based on the literature mentioned above we expect elevated yolk THs to advance the timing of puberty, gonadal regression and moult. The rate of moult should also be influenced, with birds receiving experimental TH elevation moulting faster. Previous studies have reported that gravid female three-spined sticklebacks (*Gasterosteus aculeatus*) exposed to predatory cues produced eggs with higher corticosterone (*Giesing et al., 2011*), disturbed embryonic transcriptome (*Mommer & Bell, 2014*), offspring with altered anti-predator behaviour (*Giesing et al., 2011*) and modified cortisol response in adulthood (*Mommer & Bell, 2013*). We may therefore expect elevated yolk THs to similarly induce long-term behavioural changes in response to environmental cues (i.e. photoperiod), via organising effects during the embryonic development. We also explored the effects of yolk THs on reproductive investment in females, another important fitness aspect. Finally, yolk THs may increase oxidative stress due to their stimulating effects on metabolism.

# MATERIALS AND METHODS

## Overview of the method

Japanese quails are easy to maintain in captivity, and their short generation time makes it a good model to investigate the long-term effects of maternal hormones. Rearing birds in captivity allowed us to apply a powerful within-female experimental design (i.e. knowing which chick hatched from which egg which is not feasible in field studies), thus reducing the effect of random variation among females. Moreover, studying the role of natural variation of prenatal THs in precocial species may give additional information to previous studies in altricial species. Finally, Japanese quail is a commonly used model in maternal hormone research with substantial literature available (*McNabb, Blackman & Cherry, 1985*; *McNabb, Dicken & Cherry, 1985*; *Wilson & McNabb, 1997*; *Okuliarova et al., 2011*).

We injected unincubated eggs from Japanese quails maintained in captivity with either $T_4$ or $T_3$ alone, a combination of both hormones, or a saline (control) solution. This design allowed us to explore the effects of $T_4$ and $T_3$ separately, which has not been done in

previous studies. The elevation in yolk THs remained within the natural range of this species, a crucial condition to obtain relevant results for an eco-evolutionary context. We measured traits known to be influenced by circulating and yolk THs: hatching success, age at embryonic mortality, growth, transition between life-history stages (i.e. reproductive state and moult) and oxidative stress.

## Parental generation and egg collection

The parental generation was composed of adult Japanese quails provided by Finnish private local breeders that were kept in two acclimated rooms. Twenty-four breeding pairs were formed by pairing birds from different breeders. Individuals were identified using metal leg rings. The floor was covered with 3–5 cm sawdust bedding. A hiding place, sand and calcium grit were provided. Each pair was housed in indoor aviary divided into pens of 1 m$^2$ floor area. The temperature was set to 20 °C with a 16L:8D photoperiod (light from 06.00 to 22.00). Food (Poultry complete feed, 'Kanan Paras Täysrehu', Hankkija, Finland) was provided ad libitum and water was changed every day.

Pairs were monitored every morning to collect eggs for 7 days. Eggs were individually marked (non-toxic marker), weighed and stored in a climate-controlled chamber at 15 °C and 50% relative humidity. On the last day of collection, a total of 4–8 eggs per pair were injected with a solution (see next section).

## Preparation of the solution, injection procedure and incubation

The preparation of hormone solution and the procedure of injection were based on previous studies (*Ruuskanen et al., 2016*; *Hsu et al., 2017*). In brief, crystal $T_4$ (L-thyroxine, ≥98% HPLC, CAS number 51–48–9, Sigma–Aldrich,St. Louis, MO, USA) and $T_3$ (3,3′,5-triiodo-L-thyronine, >95% HPLC, CAS number 6893-02-3, Sigma–Aldrich,St. Louis, MO, USA) were first dissolved in 0.1 M NaOH and then diluted in 0.9% NaCl. The injection of THs resulted in an increase of two standard deviations ($T_4$ = 8.9 ng/egg, equivalent to 1.79 pg/mg yolk; $T_3$ = 4.7 ng/egg, equivalent to 1.24 pg/mg yolk), a recommended procedure for hormone manipulation within the natural range (*Ruuskanen et al., 2016*; *Hsu et al., 2017*; *Podmokła, Drobniak & Rutkowska, 2018*). The control solution (CO) was a saline solution (0.9% NaCl). The concentrations of the hormone solutions were based on previous measurements of 15 eggs from the same flock (content per egg (SD) $T_4$ = 15.3 (4.4) ng, $T_3$ = 7.6 (2.3) ng; concentrations (SD), $T_4$ = 4.20 (0.89) pg/mg yolk, $T_3$ = 2.10 (0.62) pg/mg yolk).

Hormone injections were performed at room temperature in a laminar hood. Eggs were put sideways, allowing yolks to float up to the middle position. Before injection, the shell was disinfected with a cotton pad dipped in 70% EtOH. We used a 27G needle (BD Microlance™) to pierce the eggshell and then used a 0.3 ml syringe to deliver 50 μl of the respective hormone solution or control. After injection, the hole was sealed with a sterile plaster (OPSITE Flexigrid, Smith & Nephew).

In total, 158 eggs were injected and divided as follows over the treatments: $T_3$ treatment ($N$ = 39); $T_4$ treatment ($N$ = 39); $T_3+T_4$ treatment ($N$ = 40) and control, CO ($N$ = 40). To balance the genetic background of the parents and the effect of storage, each egg laid by
the same female was sequentially assigned to a different treatment and the order of treatments was rotated among females. After injection, eggs were placed in an incubator at 37.8 °C and 55% relative humidity. Until day 14 after starting incubation, eggs were automatically tilted every hour by 90°. On day 14, tilting was halted and each egg was transferred to an individual container to monitor which chick hatched from which egg. On day 16 after injection, (normal incubation time = 17 days), the temperature was set to 37.5 °C and the relative humidity to 70%. Eggs were checked for hatching every 4 h from day 16 onwards. Four days after the first egg hatched, all unhatched eggs were stored in a freezer and dissected to determine the presence of an embryo. The age of developed embryos was assessed according to *Ainsworth, Stanley & Evans (2010)*.

## Rearing conditions of the experimental birds

In total, 66 chicks hatched ($N$ = 10 CO, 15 $T_3$, 20 $T_4$ and 21 $T_3T_4$), yielding a rather low overall hatching success (ca. 40%). Among the unhatched eggs, 33.7% (31 out of 92) had no developed embryos, and these were evenly distributed between the treatments (CO = 9/40, $T_3$ = 8/39, $T_3T_4$ = 8/40 and $T_4$ = 6/39 eggs). Discarding the unfertilised eggs gives an overall hatching success of ca. 51%. Previous studies on Japanese quails have reported comparable hatching success, even in unmanipulated eggs (e.g. 40% in *Okuliarová, Škrobánek & Zeman (2007)*; ca. 60% in *Pick et al. (2016)* and in *Stier, Metcalfe & Monaghan (2019)*). In addition, the injection procedure itself is also known to reduce hatching success to some extent (*Groothuis & Von Engelhardt, 2005*). Twelve hours after hatching, the chicks were marked by a unique combination of coloured rings and nail coding and transferred to two cages of 1 m$^2$ floor area and ca. 30 cm height (ca. 30 chicks/cage, sex and treatments mixed together). The chicks were provided with heating mats and lamps as extra heat sources for the first 2 weeks. The chicks were fed with sieved commercial poultry feed ('Punaheltta paras poikanen', Hankkija, Finland), and provided with Calcium and bathing sand. A total of 2 weeks after hatching, the chicks were separated in four 1 m$^2$ cages (ca. 30 cm high) of about 16 individuals. Around 3 weeks after hatching, coloured rings were replaced with unique metal rings. On week 4 after hatching, birds were transferred to eight pens of 1 m$^2$ floor area (average of 7.1 birds/pen, range = 4–9), under the same conditions as the parents. Around the age of sexual maturity (ca. 6–8 weeks after hatching), the birds were separated by sex in twelve 1 m$^2$ pens (average of 4.8 birds/pen, range = 4–5). The chicks were under the same photoperiod as the adults (i.e. 16L:8D).

## Monitoring of growth and reproductive maturation

Body mass and wing length were measured twelve hours after hatching. Tarsus was not measured because it bends easily, resulting in inaccurate measures and potential harm for the young. From day 3 to day 15, these three traits were monitored every 3 days. From day 15 to day 78 (ca. 12 weeks), chicks were measured once a week. Body mass was recorded using a digital balance to the nearest 0.1 g. Wing and tarsus lengths were respectively measured with a ruler and a calliper to the nearest 0.5 mm and 0.1 mm. The sample size for the growth analysis was 7 CO, 11 $T_3$, 18 $T_4$ and 21 $T_3T_4$. From week 6

to week 10, we monitored cloacal gland development and foam production in 28 males. Cloacal glands were measured every other day with a calliper to the nearest 0.1 mm as a proxy for testes development and sexual maturation (*Biswas et al., 2007*). Foam production (by gently squeezing the cloacal gland) was assessed at the same time and coded from 0 (no foam) to 3 (high production of foam), as a Proxy for cloacal gland function (*Cheng, Hickman & Nichols, 1989*; *Cheng, McIntyre & Hickman, 1989*). The same observer performed all measurements. We collected eggs produced by 10-week-old females over a 6-day period and recorded their mass to the nearest 0.1 g. We collected on average 5.7 eggs (range = 4–7) per female from 28 females.

## Monitoring of cloacal gland regression and moult

In Japanese quails, exposure to short photoperiod and cold temperature triggers reproductive inhibition and postnuptial moulting (*Tsuyoshi & Wada, 1992*). THs are known to coordinate these two responses (see introduction). When the birds reached the age of ca. 7 months, we exposed them to short photoperiod (8L:16D, that is light from 08.00 to 16.00) with a 12:12-h cycle of normal (20 °C) and low (9 °C) temperature (low temperature was effective from 18.00 to 06.00). Cloacal gland regression (as a proxy for testes regression) was monitored every other day for 2 weeks with a calliper by measuring the width and length to obtain the area of the gland to the nearest 0.1 mm$^2$ ($N$ = 26 males; 4 CO, 4 $T_3$, 8 $T_4$ and 12 $T_3T_4$). Primary moult was recorded from a single wing by giving a score to each primary from 0 (old feather) to 5 (new fully-grown feather) following *Ginn & Melville (1983)* ($N$ = 54 males and females; 7 CO, 11 $T_3$, 16 $T_4$ and 20 $T_3T_4$). The total score of moulting was obtained by adding the score of all feathers.

## Oxidative status biomarker analyses

Two blood samples were drawn, when birds were 2 weeks ($N$ = 58 chicks) and 4 months old ($N$ = 55 adults), respectively. After discarding missing values, the sample size per treatment and age class was 6 CO, 10 $T_3$, 17 $T_4$ and 18 $T_3T_4$ chicks, and 6 CO, 10 $T_3$, 15 $T_4$ and 18 $T_3T_4$ adults. 200 µl of blood was collected from the brachial vein in heparinised capillaries and directly frozen in liquid nitrogen. Then, the samples were stored at −80 °C until analyses. We measured various biomarkers of antioxidant status; the antioxidant glutathione (tGSH), the ratio of reduced and oxidised glutathione (GSH:GSSG) and activity of the antioxidant enzymes glutathione peroxidase (GPx), catalase (CAT) and superoxide dismutase (SOD) from the blood. Measuring multiple biomarkers of oxidative and antioxidant status allows a broader understanding of the mechanism, and the interpretation of the results is more reliable if multiple markers show similar patterns. The GSH:GSSG ratio represents the overall oxidative state of cells and a low ratio reveals oxidative stress (*Hoffman, 2002*; *Isaksson et al., 2005*; *Lilley et al., 2013*; *Rainio et al., 2013*; *Halliwell & Gutteridge, 2015*). GPx enzymes catalyse the glutathione cycle, whereas CAT and SOD directly regulate the level of reactive oxygen species (ROS) (*Ercal, Gurer-Orhan & Aykin-Burns, 2001*; *Halliwell & Gutteridge, 2015*). The methodology for measuring each biomarker is described in detail in *Rainio et al. (2015)*. All analyses were conducted blindly of the treatment following *Ruuskanen et al. (2017)*.

## Ethics

The study complied with Finnish regulation and was approved by the Finnish Animal Experiment Board (ESAVI/1018/04.10.07/2016). In case of signs of harassment or disease, birds were placed in quarantine and monitored daily until they had recovered. Criteria for humane endpoints were defined as follow: passive behaviour, loss of appetite, loss of 30% of body weight, moving abnormally, trouble breathing. If we observed no clear improvement after two days, we would consult the veterinarian. A bird would be euthanised if it does not show signs of improvement in the next two days, though some judgement can be applied based on the alleged cause. One male was euthanised before the end of the experiment due to severe head injury. At the end of the experiment, all birds were euthanised by decapitation for collection of tissue samples (not used in this study).

## Statistical analysis

Data were analysed with the software R version 3.5.3 (*R Core Team, 2019*). In this study, two different statistical approaches were used: null-hypothesis testing with Generalised Linear Mixed Models (GLMMs) and Linear Mixed Models (LMMs), and multimodel inference with Generalised Additive Mixed Models (GAMMs). GAMMs were used to analyse the data on body and cloacal gland growth to account for its non-linear pattern (see Growth). In this analysis, we preferred multimodel inference as GAMMs generate many candidate models that cannot be directly compared (e.g. by the Kenward–Roger approach). Instead, candidate models were ranked based on their Akaike Information Criterion (AIC) values. Models with a $\Delta$AIC $\leq 2$ from the top-ranked model were retained in the set of best models. Akaike weights of all models were calculated following (*Burnham & Anderson, 2002*), and evidence ratios of the top-ranked models were calculated as the weight of a model divided by the weight of the null model (*Burnham, Anderson & Huyvaert, 2011*). To estimate the effect of the predictors, we computed the 95% confidence intervals from the best models using the *nlme* package (*Pinheiro et al., 2018*). GLMMs and LMMs were fitted using the R package *lme4* (*Bates et al., 2015*), and GAMMs were fitted using the package *mgcv* (*Wood, 2017*). *P*-values for GLMMs were obtained by parametric bootstrapping with 1,000 simulations and *p*-values for LMMs were calculated by model comparison using Kenward–Roger approximation, using the package *pbkrtest* in both cases (*Halekoh & Højsgaard, 2014*). Post-hoc Tukey analyses were conducted with the package *multcomp* (*Hothorn, Bretz & Westfall, 2008*). Model residuals were checked visually for normality and homoscedasticity. Covariates and interactions were removed when non-significant ($\alpha = 0.05$).

Effect size calculations (Cohen's *d* and 95% CI) were performed with the website estimationstats.com (*Ho et al., 2019*) and statistical power analyses were performed using *t*-tests for independent means with GPower (*Faul et al., 2009*) with the effect size values calculated. When presenting and discussing our results, we use the language of statistical 'clarity' rather than statistical 'significance' as suggested by *Dushoff, Kain & Bolker (2019)*.

### Hatching success

To analyse hatching success, each egg was given a binary score: 0 for unhatched egg and 1 for hatched egg. A GLMM was fitted with a binomial error distribution (logit link) and mother identity as a random intercept and the 4-level treatment as the predictor. Egg mass might affect hatchability and was therefore added as a covariate in both models.
The potential effect of storage duration on hatchability (*Reis, Gama & Soares, 1997*) was accounted for by including laying order as a covariate in both models. This covariate allowed us to control for the age of the egg as well.

### Duration of embryonic period, age at embryonic mortality and early morphological traits

Duration of embryonic period and early morphological traits (mass and wing length at hatching, and tarsus length at day 3) were modelled with separate LMMs. Treatment, sex of the individuals and egg mass were included as fixed factors. Laying order was added as a covariate to account for potential effects of storage duration on hatching time and on chick weight (*Reis, Gama & Soares, 1997*). Mother identity was included as a random intercept.

The data for embryonic age had a skewed distribution and residuals were not normally distributed and heterogenous, which violated LMM assumptions on residual distribution. We therefore performed a simple Kruskal–Wallis test.

### Growth

As growth curves typically reach an asymptote, we fitted non-linear GAMMs to these curves. Growth in body mass, tarsus and wing length were analysed in separate GAMMs. Growth was analysed until week 10 after hatching as all birds appeared to have reached their maximum body mass and tarsus and wing length. The data are composed of repeated measurements of the same individuals over time; therefore, we first corrected for temporal autocorrelation between the measurements using an ARMA (1,1) model for the residuals (*Zuur et al., 2009*). Second, as mothers produced several eggs, the models included nested random effects, with measured individuals nested into mother identity, allowing for random intercepts. GAMMs allow modelling the vertical shift of the curves (i.e. changes in intercepts) and their shape. Treatment and sex were included as predictors. A smoothing function for the age of the birds was included to model the changes in the growth curves, and was allowed to vary by sex or treatment only, or none of these predictors. The interaction between sex and treatment was not analysed due to low statistical power. Additive effect of treatment and sex was tested for the intercept but could not be computed for curve shape. All combinations of the relevant predictors were tested for both shape parameters (i.e. intercept and curve shape).

Prenatal THs may exert most of their effects in the offspring early life; this is why we additionally tested hatchlings morphological traits apart from the growth curve. Likewise, we also analysed separately morphological traits at adulthood (ca. 9 weeks old),

**Table 1 Loadings of the different antioxidant biomarkers on the principal components 1 and 2.**

| Factor loadings | PC1 (34.0%) | PC2 (26.2%) |
|---|---|---|
| CAT | −0.49 | 0.14 |
| SOD | 0.20 | −0.71 |
| GST | −0.65 | −0.10 |
| GP | 0.04 | −0.63 |
| tGSH | −0.60 | −0.26 |

as these traits may condition the fitness of an individual. Because of sex differences and low sex-specific sample sizes, we standardised the measures within sex and regressed the standardised responses against treatment in a linear regression.

### Reproductive maturation, regression and investment

Due to low sample sizes in these sex-specific responses, we could not perform robust statistical analyses. We therefore present these analyses and results in the Supplemental Material and only briefly discuss them (Figs. S6–S9; Table S3).

### Moult

Two parameters of moult were analysed in separate LMMs: the timing of moult (i.e. the moult score after 1 week of short photoperiod), and the rate of moult (i.e. how fast birds moulted). Both models included treatment and sex as fixed factors, and mother identity as a random intercept. The rate of moult was tested by fitting an interaction between treatment and age. This model also included the main effect of age and individual identity, nested within mother identity, as a random intercept to account for repeated measures. Estimated marginal means and standard errors (EMMs ± SE) were derived from the model using the package *emmeans* (*Lenth, 2019*).

### Oxidative stress

A principal component analysis (PCA) was first performed on measured antioxidant markers (SOD, CAT, GPx, tGSH and GST), to reduce the number of metrics for subsequent analyses. The first and the second principal components (PCs) explained together 60.2% of the variance (Table 1). PC1 and PC2 were then used as dependent variables in separate LMMs. LMMs included the treatment, sex and age of individuals (2 weeks and 4 months old) as fixed factors and the 2-way interactions between treatment and sex, and treatment and age. Mother and individual identities, to account for repeated measures, were added as random intercepts. Malondialdehyde (MDA) is a marker of oxidative damage, which is a different measure from antioxidant activity, and was therefore analysed in a separate LMM using the same parameters as for PC1 and PC2, adding the batch of the assay as an additional random intercept. The marker of cell oxidative status (GSH:GSSG ratio) was analysed with the same model used for PC1 and PC2.

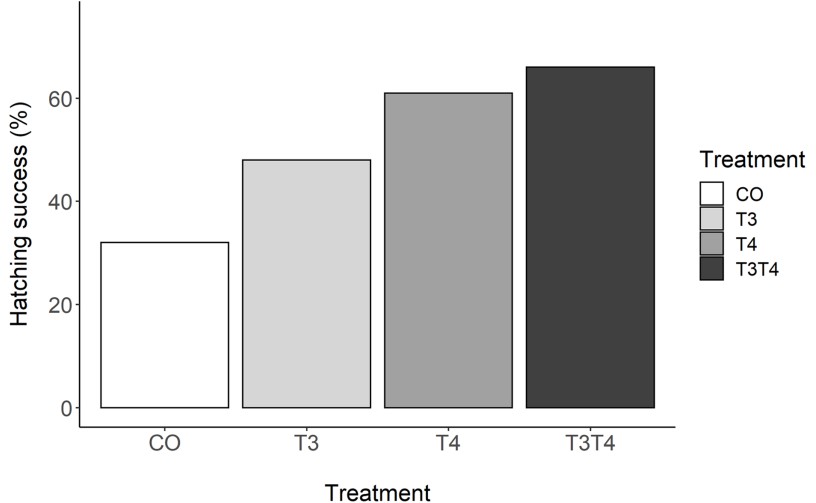

**Figure 1  Percentage of hatching success according to yolk TH manipulation treatments.** $N$ = 40 CO, 39 $T_3$, 39 $T_4$ and 40 $T_3T_4$. CO = control, $T_4$ (thyroxine) = injection of $T_4$, $T_3$ (triiodothyronine) = injection of $T_3$, $T_3T_4$ = injection of $T_3$ and $T_4$.  

## RESULTS

### Effects of prenatal THs on hatching success and age of embryo mortality

There was a clear effect of elevated prenatal THs on hatching success (GLMM, $p = 0.03$, Fig. 1). Tukey post-hoc analysis revealed that hatching success in the $T_3T_4$ (66%) group was statistically higher than in the CO group (32%) (Tukey $z = 2.77$, $p = 0.03$). There was a non-significant trend between the $T_4$ (61%) and the CO groups ($z = 2.37$, $p = 0.08$). There were no clear differences in hatching success between the $T_3$ (48%) and the CO group ($z = 1.25$, $p = 0.45$), or between the hormone treatments (all $z < 1.61$, all $p > 0.37$).

Dissection of the unhatched eggs showed that age of embryo mortality did not differ between the treatments (Kruskal–Wallis $\chi^2 = 7.22$, df = 3, $p = 0.07$; Fig. S1). Finally, the manipulation of yolk THs did not affect the duration of embryonic period (LMM, $F_{3,42.0} = 0.57$, $p = 0.64$, Fig. S2). Sex of the embryo or egg mass (LMM sex, $F_{1,49.7} = 2.63$, $p = 0.11$; LMM egg mass, $F_{1,19.3} = 0.01$, $p = 0.92$) were also not associated with the duration of the embryonic period. Laying order (i.e. the effect of storage duration) was not correlated with any of the responses (all $p \geq 0.25$).

### Effects of prenatal THs on growth

Mass at hatching was not influenced by the elevation of prenatal THs (LMM, $F_{3,35.0} = 0.81$, $p = 0.50$, Fig. S3). Mass at hatching was positively correlated with egg mass (LMM, Estimate ± SE = 0.72 ± 0.10 g, $F_{1,24.1} = 46.9$, $p < 0.001$). Although we detected no clear differences on hatchling morphological traits (body mass, wing and tarsus length) due to prenatal THs (all $p > 0.12$), the calculated effect sizes (Cohen's $d$ (95% CI)) and achieved statistical power yielded additional information regarding the potential effects of prenatal THs (Table 2). For body mass, the effect sizes were low and the achieved statistical power was very low. For wing length, the effect sizes were moderate and the achieved

**Table 2 Cohen's *d*, 95% CIs and achieved statistical power for post-hatching and adult morphological measures (body mass, wing and tarsus length).**

| Contrast | Hatchlings | | | Adults | | |
|---|---|---|---|---|---|---|
| | Cohen's *d* | 95% CI | Statistical power (1-β) | Cohen's *d* | 95% CI | Statistical power (1-β) |
| Body mass | | | | | | |
| CO-T3 | 0.22 | [−0.70 to 1.09] | 0.13 | −0.01 | [−1.23 to 1.05] | 0.05 |
| CO-T4 | 0.17 | [−0.79 to 1.11] | 0.11 | −0.46 | [−1.54 to 0.57] | 0.26 |
| CO-T3T4 | 0.06 | [−1.03 to 1.05] | 0.07 | −0.48 | [−1.61 to 0.48] | 0.28 |
| T3-T4 | −0.09 | [−0.81 to 0.66] | 0.08 | −0.51 | [−1.32 to 0.22] | 0.36 |
| T3-T3T4 | −0.24 | [−1.0 to 0.52] | 0.17 | −0.53 | [−1.28 to 0.19] | 0.40 |
| T4-T3T4 | −0.16 | [−0.80 to 0.49] | 0.13 | −0.01 | [−0.66 to 0.65] | 0.05 |
| Wing length | | | | | | |
| CO-T3 | −0.41 | [−1.33 to 0.36] | 0.23 | −1.07 | [−2.09 to 0.36] | 0.67 |
| CO-T4 | −0.79 | [−1.62 to 0.10] | 0.57 | −0.72 | [−1.52 to 0.20] | 0.47 |
| CO-T3T4 | −0.56 | [−1.31 to 0.13] | 0.37 | −0.81 | [−1.52 to 0.11] | 0.56 |
| T3-T4 | −0.41 | [−1.09 to 0.31] | 0.31 | 0.12 | [−0.54 to 0.91] | 0.09 |
| T3-T3T4 | −0.20 | [−0.80 to 0.54] | 0.14 | 0.09 | [−0.49 to 0.91] | 0.08 |
| T4-T3T4 | 0.19 | [−0.44 to 0.86] | 0.15 | −0.03 | [−0.67 to 0.62] | 0.06 |
| Tarsus length | | | | | | |
| CO-T3 | −0.58 | [−1.29 to 0.33] | 0.33 | −0.10 | [−1.25 to 1.08] | 0.07 |
| CO-T4 | −0.88 | [−1.93 to 0.08] | 0.61 | −0.67 | [−1.59 to 0.31] | 0.43 |
| CO-T3T4 | −0.92 | [−2.02 to 0.06] | 0.68 | −0.78 | [−1.73 to 0.09] | 0.54 |
| T3-T4 | 0.01 | [−0.91 to 1.02] | 0.05 | −0.68 | [−1.41 to 0.08] | 0.53 |
| T3-T3T4 | 0.05 | [−0.91 to 1.07] | 0.06 | −0.79 | [−1.44 to -0.11] | 0.67 |
| T4-T3T4 | 0.06 | [−0.69 to 0.79] | 0.07 | −0.11 | [−0.73 to 0.54] | 0.10 |

Note:
95% CIs were calculated by bootstrap resampling with 5,000 resamples. CO = control, $T_4$ (thyroxine) = injection of $T_4$, $T_3$ (triiodothyronine) = injection of $T_3$, $T_3T_4$ = injection of $T_3$ and $T_4$.

statistical power was low. For tarsus length, the effect sizes were moderate to large and the achieved statistical power was low to moderate. Similarly, adult morphology was not affected by the treatment (all $p > 0.13$), but effect sizes indicate small to large effects of prenatal THs (Table 2). For body mass, the effect sizes were small and the achieved power was low. For wing length, the effect sizes were large and the achieved power was moderate. For tarsus length, the effect sizes were small to large and the achieved power was moderate to high.

Regarding body mass growth, the top-ranked model showed that the curve shape and the intercept differ according to sex (Table 3). After 10 weeks, females had a larger body mass than males (mean ± SE females = 214.4 ± 5.7 g, males = 172.4 ± 4.5 g, Fig. 2), which was supported by the 95% CIs (Table 4). Based on model selection we conclude that the treatment had no effect on body mass growth (Table 3).

For wing growth, the top-ranked model (ΔAIC ≤ 2) included sex in the intercept, while treatment was not included in the best supported model (Table S1). The 95% CIs (Table 3) confirmed that males had a lower wing length than females (Fig. S4).

**Table 3 Results of the Generalised Additive Mixed Models (GAMMs) on body mass growth, with sex and treatment fitted either as intercept, curve shape or both (all combinations tested).**

| Model | Intercept | Curve shape | ΔAIC | df | Weight |
|---|---|---|---|---|---|
| 1 | Sex | Sex | 0.0 | 11 | 0.8430 |
| 8 | Treatment + sex | Sex | 3.5 | 14 | 0.1497 |
| 3 | – | Sex | 9.9 | 10 | 0.0061 |
| 2 | Treatment | Sex | 13.2 | 13 | 0.0012 |
| 11 | Sex | – | 77.6 | 9 | <0.001 |
| 9 | Treatment + sex | – | 81.6 | 12 | <0.001 |
| 12 | – | – | 91.2 | 8 | <0.001 |
| 10 | Treatment | – | 95.0 | 11 | <0.001 |
| 5 | Sex | Treatment | 147.9 | 15 | <0.001 |
| 7 | Treatment + sex | Treatment | 151.7 | 18 | <0.001 |
| 6 | – | Treatment | 161.2 | 14 | <0.001 |
| 4 | Treatment | Treatment | 165.5 | 17 | <0.001 |

**Note:**
A total of 12 GAMMs were fitted and ranked based on their AIC, from the lowest to the highest. Weight: Akaike's weight.

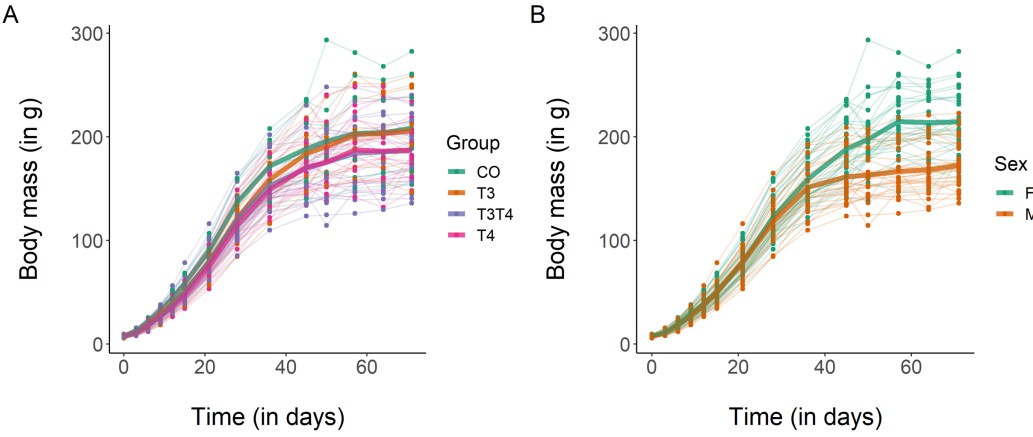

**Figure 2 Growth curves in body mass of Japanese quails hatching from eggs treated with either $T_3$, $T_4$, a combination of both hormones, or a control solution.** See Fig. 1 for a description of the treatments. Each line represents an individual bird, while thick coloured lines represent mean values. (A) Growth curve according to yolk TH manipulation. $N = 7$ CO, 11 $T_3$, 18 $T_4$ and 21 $T_3T_4$. (B) Growth curve according to sex. $N = 29$ females and 28 males.

Concerning tarsus growth, the models within ΔAIC ≤ 2 included no predictors for the curve shape but included treatment for the intercept (Table S2). The 95% CIs of the parameter estimates from these models suggested that there was a slight negative effect of $T_3T_4$ treatment on tarsus growth (Table 4; Fig. S5). However, as the estimates were close to 0 (Table 4) and evidence ratios showed that the model with treatment as a predictor was only 3.5 times more supported than the null model (Table S2), we conclude that the effect of THs on tarsus length is likely to be very small. Likewise, the second model for tarsus length included sex as a predictor for the intercept, but its 95% CIs overlapped with 0 (Table 4). We therefore conclude that sex had no clear effect on tarsus growth.

**Table 4 95% confidence intervals of the predictors in the top-ranked models according to AIC values (see Table 2; Tables S1 and S2).**

| Curve parameter | Predictors | Lower limit | Estimate | Upper limit |
|---|---|---|---|---|
| (A) Body mass (model 1) | | | | |
| Intercept | **Sex (M)** | −19.7 | −12.6 | −5.5 |
| Curve shape | **Sex (F)** | 9.9 | 20.0 | 30.0 |
| Curve shape | **Sex (M)** | 14.3 | 24.5 | 34.7 |
| (B) Wing length (model 11) | | | | |
| Intercept | **Sex (M)** | −2.3 | −1.2 | −0.1 |
| Curve shape | **Age** | 26.4 | 28.7 | 31.0 |
| (C) Tarsus length (model 10) | | | | |
| Intercept | Treatment ($T_3$) | −0.8 | 0.02 | 0.8 |
| Intercept | **Treatment ($T_3T_4$)** | −1.5 | −0.8 | −0.1 |
| Intercept | Treatment ($T_4$) | −1.3 | −0.6 | 0.2 |
| Curve shape | **Age** | 10.5 | 11.1 | 11.8 |
| Tarsus length (model 9) | | | | |
| Intercept | Treatment ($T_3$) | −0.9 | −0.07 | 0.7 |
| Intercept | **Treatment ($T_3T_4$)** | −1.5 | −0.8 | −0.1 |
| Intercept | Treatment ($T_4$) | −1.4 | −0.6 | 0.1 |
| Intercept | Sex (M) | −0.8 | −0.3 | 0.3 |
| Curve shape | **Age** | 10.5 | 11.1 | 11.7 |

**Note:**
Predictors in bold have confidence intervals that do not overlap with 0. For the intercept, the reference groups are female and CO for the predictors sex and treatment, respectively.

## Effects of prenatal THs on postnuptial moult

As expected, birds started to moult soon after being exposed to short photoperiod, with an average increase of moult score by 6 per week (SE = 0.2, $F_{1,254.0} = 827.4$, $p < 0.001$, Fig. 3). The first moult score (assessed one week after switching to short photoperiod) was not affected by the treatment (LMM, $F_{3,42.7} = 0.36$, $p = 0.78$), but was influenced by sex, with females having a higher score than male (EMMs ± SE: female = 21.4 ± 1.6, male = 7.2 ± 1.7; LMM $F_{1,45.3} = 41.9$, $p < 0.001$). Yolk TH elevation did not affect the rate of moult (LMM interaction treatment × time, $F_{3,251.0} = 0.59$, $p = 0.62$, Fig. 3).

## Effects of prenatal THs on oxidative stress

The elevation of yolk THs had no effect on PC1 or PC2 of antioxidants at either 2 weeks ('chicks') or 4 months ('adults') old (LMM on PC1, $F_{3,40.3} = 2.40$, $p = 0.08$; LMM on PC2, $F_{3,42.2} = 0.92$, $p = 0.44$, treatment × age, $F < 0.91$, $p > 0.44$). The age of the birds had a highly significant effect on PC1, with chicks generally having higher antioxidant capacities (CAT, GST and tGSH) than adults (LMM, Estimate ± SE = −1.34 ± 0.19, $F_{1,49.2} = 52.1$, $p < 0.001$). All the other predictors had no effect on either PC1 or PC2 (all $F < 2.93$ and all $p > 0.09$).

The marker of oxidative damage, MDA, was affected by the elevation of yolk THs (LMM, $F_{3,43.6} = 3.08$, $p = 0.04$, Fig. 4). Tukey post-hoc analysis showed that the $T_4$ group had higher MDA values than the $T_3$ group (Estimate ± SE = 0.01 ± 0.004, Tukey contrast

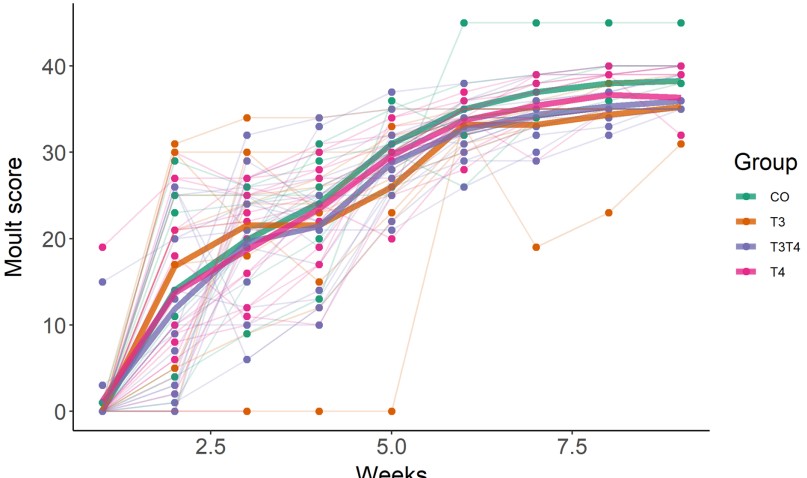

**Figure 3 Primary moult score in 7-month old Japanese quails according to yolk TH manipulation treatments.** $N$ = 7 CO, 11 $T_3$, 16 $T_4$ and 20 $T_3T_4$. See Fig. 1 for a description of the treatments. Measures were taken once a week after switching from long photoperiod (16L:8D) to short photoperiod (8L:16D, switch = time point 0 on $x$-axis). Each line represents an individual bird, while thick coloured lines represent group mean values.

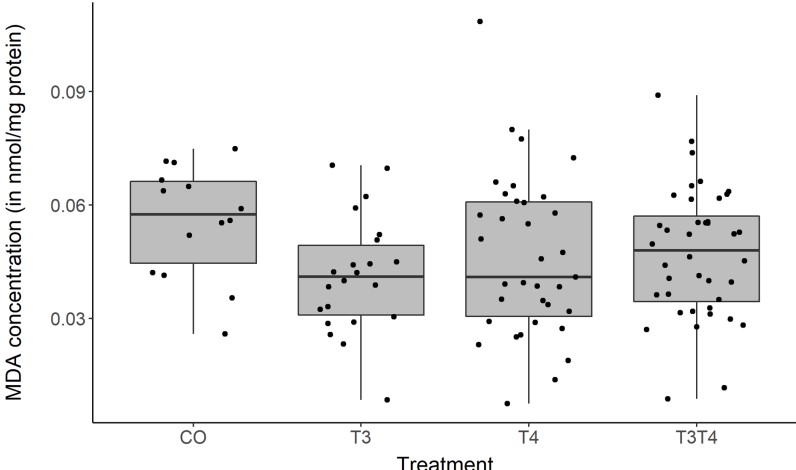

**Figure 4 MDA concentration according to yolk TH manipulation treatments.** Samples from two ages pooled: $N$ = 7 CO, 11 $T_3$, 17 $T_4$ and 20 $T_3T_4$. See Fig. 1 for a description of the treatments. Boxplots show median and quartiles.

$p$ = 0.01), but none of the groups differed from the control (Tukey $p$-values > 0.19). However, this result became non-significant when removing the outlier in the $T_4$ group (LMM, $F_{3,43.1}$ = 2.68, $p$ = 0.06). MDA levels were not affected by the age or the sex of individuals (LMM age, $F_{1,54.4}$ = 0.30, $p$ = 0.59; LMM sex, $F_{1,42.0}$ = 1.47, $p$ = 0.23).

The marker of cell oxidative balance, GSH:GSSG, was not influenced by yolk THs nor by the sex of the birds (LMM treatment, $F_{3,33.0}$ = 0.85, $p$ = 0.48; LMM sex, $F_{1,40.6}$ = 0.57, $p$ = 0.45). However, chicks had a higher GSH:GSSG ratio than adults (LMM, Estimate ± SE = 0.17 ± 0.04, $F_{1,50.0}$ = 18.3, $p$ < 0.001).

## DISCUSSION

The aim of this experimental study was to investigate the potential short-term and organisational effects (with long-term consequences) of maternal THs in a precocial species, the Japanese quail, by experimental elevation of THs in eggs. Our study is the first to investigate the effects of yolk $T_3$ and $T_4$ separately, within the natural range of the study model. In addition we studied both short-and long-term effects on embryonic development, growth, life stage transitions and oxidative stress. We detected a positive effect of yolk THs on hatching success. All other response variables studied were not clearly affected by elevated prenatal THs.

### Effects of prenatal THs on hatching success and embryonic development

The overall low hatching success, and especially in the control group, forces us to interpret these results with caution. In addition, we cannot exclude that our results may be partly due to selective disappearance of lower quality embryos in the control group and with injected THs helping lower quality chicks to hatch. This might have biased the results after hatching, but is still a relevant effect of the hormone treatment. We found that hatching success almost doubled when the eggs received an injection of both $T_4$ and $T_3$, or an injection of $T_4$ only. Previous similar studies reported comparable effects of yolk THs in rock pigeons (*Hsu et al., 2017*) and in collared flycatchers (*Hsu et al., 2019*). In these studies, injections consisted of a mixture of both $T_3$ and $T_4$. Given that mostly $T_3$ binds to receptors, these results suggest that embryos likely express deiodinase enzymes to convert $T_4$ to $T_3$, and/or yolk may contain maternally derived deiodinase mRNA, as injection with $T_3$ only did not differ from control. Indeed, deiodinase expression has previously been characterised in chicken embryos already 24h after the onset of incubation (*Darras et al., 2009*). An old study found that injecting $T_4$ close to hatching can advance hatching time, which suggests that yolk THs may help embryos overcoming hurdles close to hatching (*Balaban & Hill, 1971*). In contrast with our study, two similar studies in altricial species detected no increased hatching success due to the injection of THs (*Ruuskanen et al., 2016*; *Sarraude et al., 2020*). The dissimilarities between the studies may come from inter-specific differences in terms of utilisation of yolk THs by the embryos or from context-dependent effects (e.g. due to other egg components). Further comparative and mechanistic studies could help understanding the dynamic of yolk THs during incubation.

Increased yolk THs did not influence age of embryo mortality. Similar to our study, *Ruuskanen et al. (2016)* did not find any difference in the timing of mortality in great tit embryos. Conversely, the study on rock pigeons found that yolk THs increased the proportion of well-developed embryos (*Hsu et al., 2017*). Similar to our result on hatching success, yolk TH effects on embryonic development may differ in a species-specific manner.

Our results on hatching success may partly be attributed to yolk THs balancing the negative effects of injections on embryonic survivability. Further studies may aim at understanding the contribution of THs to counteract the effect of injection. To do so, such
studies may use an non-invasive method to manipulate yolk THs (e.g. egg-dipping method as in *Perrin et al. (1995)*), in addition to injected controls, like in our study.

## Effects of prenatal THs on growth

We found no apparent influence of yolk THs on growth, contrary to our expectations based on the recent literature. Other comparable studies found either a positive (*Hsu et al., 2019*; weak effect in *Sarraude et al. (2020)*), a negative (*Hsu et al., 2017*) or a sex-specific effect (*Ruuskanen et al., 2016*) of yolk THs on growth. This notable difference may be due to the captive conditions experienced by the Japanese quails in our study, with unrestricted access to food and water. Although the pigeon study also provided ad libitum food, parents still needed to process food before feeding their nestlings in the form of crop milk, whereas precocial quails have no such limitation. In addition, the Japanese quail has been domesticated for many generations, and probably selected for rapid growth for economic reasons. Whole-genome sequencing in chickens showed that domestication induced a strong positive selection on genes associated with growth (*Rubin et al., 2010*). Interestingly, that study also found a strong selection for a locus associated with thyroid stimulating hormone (TSH) receptor. TSH controls most of the TH production by the thyroid gland (*McNabb & Darras, 2015*), and this artificial selection may overshadow the effects of natural variations of prenatal THs on growth. Besides, the low number of individuals in the control and $T_3$ groups (7 and 11, respectively) limited the statistical power to detect differences between all the treatments. Indeed, we were able to detect small to moderate negative effects of yolk THs on morphological traits at hatching and in adulthood. Such negative effects, although small, may still be biologically relevant. Repeating the study with a larger sample size may allow us to ascertain the effects of yolk THs on growth in precocial study models. Research on the influences of prenatal THs on growth will also benefit from experimental studies on wild precocial species.

## Effects of prenatal THs on postnuptial moult

Short photoperiod in combination with cold temperature triggered primary moult, as expected. However, we detected no effect of yolk THs on the timing or speed of moult. THs are important in moult and feather growth (reviewed in *Dawson (2015)*). For example thyroidectomised birds fail to moult after being exposed to long photoperiods (*Dawson, 2015*). In addition, thyroidectomised nestling starlings failed to grow normal adult plumage and grown feathers presented an abnormal structure (*Dawson et al., 1994*). By removing the thyroid gland, these two studies implemented extreme pharmacological protocols that differ drastically from our injection of physiological doses. In addition, our experimental design, increasing TH exposure (vs decreased TH exposure in the above-mentioned studies), may have different consequences. For example there may be a threshold above which any additional hormones may not affect moult.

Overall, our results show no support for the hypothesis of organising effect of prenatal THs on life stage transitions. Yet, due to small sample sizes in sex-specific analyses (i.e. male gonadal maturation and regression and female reproductive investment), there remains a relatively high uncertainty about the potential organising effects of prenatal

THs. Replicate studies with larger samples sizes and different study models will reduce this uncertainty.

### Effects of prenatal THs on oxidative stress

In contrast to our predictions, elevated yolk THs did not affect oxidative status during chick or adult phase. We found no changes in antioxidant activities in relation to yolk THs and no imbalance in the oxidative cell status. Nevertheless, $T_4$ birds had a higher level of oxidative damage on lipids than $T_3$ birds, but this was a weak effect driven by one outlier. The lack of effects on chick oxidative status among the treatment groups could be explained by the absence of treatment effects on growth, given that high growth rates usually result in higher oxidative stress and damage (*Alonso-Alvarez et al., 2007*). In turn, the lack of treatment effects on adult oxidative status may suggest no organisational effects of prenatal THs on adult metabolism. Two recent studies in altricial species also found no influence of yolk THs on nestling oxidative stress (*Hsu et al., 2019*; *Sarraude et al., 2020*), yet telomere length, a biomarker of aging was affected (*Stier et al., 2020*). Our study shows for the first time that prenatal THs have no influence on adult oxidative stress either. The previous study focused on a limited set of biomarkers: one antioxidant enzyme, oxidative damage on lipids and oxidative balance. In the present study, we measured seven biomarkers, thus providing broader support to the absence of effects of prenatal THs on post-natal/hatching oxidative stress.

## CONCLUSION

To our knowledge, this study is the first one to experimentally investigate the consequences of natural variations of maternal THs not only early but also in adult physiology and postnuptial moult in any vertebrate. Furthermore, this study explored for the first time the effects of maternal $T_3$ and $T_4$ separately. We found no evidence for differential effects of maternal $T_4$ and $T_3$, while an effect of $T_4$, alone or in combination with $T_3$, on hatching success suggests that $T_4$ is converted into $T_3$, the biologically active form during embryonic development. Contrary to similar studies on wild altricial species, we found no influence of maternal THs on growth. Further research on embryos utilisation of maternal THs may help understand the differences observed between precocial and altricial species. Studies in other vertebrates are urgently needed to understand the potential organising effects of maternal THs with long-term consequences.

## LIST OF SYMBOLS AND ABBREVIATIONS

**CAT**      catalase
**CO**      control treatment
**GP**      glutathione peroxidase
**tGSH**      oxidised glutathione
**GSSG**      reduced glutathione
**GST**      Glutathione S-transferase
**MDA**      malonaldehyde
**SOD**      super-oxide dismutase

| $T_3$ | triiodothyronine |
| $T_4$ | thyroxine |
| THs | thyroid hormones |

## ACKNOWLEDGEMENTS

We thank Sophie Michon for her help on setting up the parental generation. We also thank Ido Pen for consultation and help with statistical analysis, and Esther Chang for her help throughout the writing phase.

### Funding

The study was funded by the Academy of Finland (grant no. 286278 to Suvi Ruuskanen), the Finnish National Agency for Education (grant no. TM-15-9960 to Tom Sarraude) and the University of Groningen (grant to Ton Groothuis). The funders had no role in study design, data collection and analysis, decision to publish, or preparation of the manuscript.

### Grant Disclosures

The following grant information was disclosed by the authors:
Academy of Finland: 286278.
Finnish National Agency for Education: TM-15-9960.
University of Groningen: Ton Groothuis.

### Competing Interests

The authors declare that they have no competing interests.

### Author Contributions

- Tom Sarraude conceived and designed the experiments, performed the experiments, analyzed the data, prepared figures and/or tables, authored or reviewed drafts of the paper, and approved the final draft.
- Bin-Yan Hsu conceived and designed the experiments, performed the experiments, analyzed the data, authored or reviewed drafts of the paper, and approved the final draft.
- Ton Groothuis analyzed the data, authored or reviewed drafts of the paper, and approved the final draft.
- Suvi Ruuskanen conceived and designed the experiments, performed the experiments, analyzed the data, authored or reviewed drafts of the paper, and approved the final draft.

### Animal Ethics

The following information was supplied relating to ethical approvals (i.e. approving body and any reference numbers):

The study complied with Finnish regulation and was approved by the Finnish Animal Experiment Board (ESAVI/1018/04.10.07/2016).

## Data Availability

Data is available at Zenodo: Sarraude, Tom, Hsu, Bin-Yan, Groothuis, Ton, & Ruuskanen, Suvi. (2020). Dataset of prenatal thyroid hormones manipulation in Japanese quails (Data set). Zenodo. DOI 10.5281/zenodo.3741711.

## Supplemental Information

Supplemental information for this article can be found online at http://dx.doi.org/10.7717/peerj.10175#supplemental-information.

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
