# Peer review of "Testing the short-and long-term effects of elevated prenatal exposure to different forms of thyroid hormones"

_PeerJ, doi:10.7717/peerj.10175_

## Round 0.1 · original submission · Major Revisions

As you will see both reviewers provided thorough and constructive reviews of your manuscript. They thought the study addresses an important area of avian development. However, they had some concerned that I would encourage you to address fully. Specifically, issues regarding hatching success and unequal sample sizes between groups.

Reviewer 1 ·

Basic reporting

The manuscript is well written and has the correct structure. It is a well written manuscript with what I find a very interesting topic. They provide good context for the study. All figures and tables are well done, although they should note what the box plots are showing (median and quartiles?). I appreciate the inclusion of all data points along with the mean (median) values on the figures.

Experimental design

The experiments are within the scope of the journal. The research question is well defined and is meaningful. Our understanding of the influence of maternally derived thyroid hormones on development in birds is incomplete and this study attempts to address this by supplementing the yolk with additional thyroid hormones. The methods are all well-defined. I wonder how effective the dosing of the yolk is for actually delivering the hormone to the developing embryo - see comments below.

Validity of the findings

I hate to say this, as I really like the idea of the experiment, but given the low hatching success of the control group I tend to agree with the previous reviewers of an earlier version of this manuscript. It would be nice for them to repeat the hatching success portion of the experiment from initial exposure through hatching to confirm their findings on hatching success, especially given this is the only parameter where they found a significant response. Your control group has a hatching success of only 25% and you are comparing it to 60% in other “control” studies. This means your control eggs had 2 to 3 times worse hatching success than other control studies. How does taking into account those eggs with no development change the hatching success of the control?

I would like to see the data on mortality. When did the embryos die during development?

Additional comments

A major question I have is in regard to the dosing of the yolk with TH. I know that these authors have used this method in the past with other species. While I appreciate that the authors are attempting to elevate TH within a somewhat natural range, I wonder how well injecting in the yolk is as a delivery method. Given this method and the lack of a response, we really don’t know when the excess TH are actually taken up by the embryo. Are the embryos actually experiencing “higher” levels of TH during development? Do they become diluted in the yolk, so that it is a very minimal dosing or does it stay concentrated and they take it up all at once? Is there a study that has injected an inert dye into the yolk and watched how it diffuses and moves through development (not suggesting you do this, just wondering if you know what happens once it gets in the yolk)? Can you provide the concentration in terms of ng T3 per ml (or mg) yolk?

Line 28: “These results suggest that yolk thyroid hormones are important in the embryonic stage of precocial birds…”

Line 60 – change “fasten” to accelerate

Line 179 - I am a little worried about the low hatching success in this study, as noted above. While they show provide references of studies where others have a 50% hatching success, there are other studies with hatching successes between 80-95% for fertilized eggs (for one see Martin and Arnold, 1991, The Condor). Additionally, their control eggs had a 25% hatching success, which is almost half of the worst hatching success they reference. It would be nice to see the distribution of ages at which the embyros died. What was the distribution between the treatments of the 31 eggs with no developed embryo? If a larger majority of these eggs were in the control, this might skew your hatching success rates away from the controls. Is the hatch rate you present in Figure 1 including the total number of eggs set or just those that showed developed embryos (discarded the infertile and those whose yolks may have been damaged resulting in no development) as the denominator? I would argue that it should be not include the infertile eggs that did not have a developed embryo. How far along were the additional 61 animals that did not hatch? I would recommend you repeat this aspect of the study to confirm the findings.

Line 291 – I applaud your use of a measure of effect sizes as a means to determine how large a response in the data is. I suggest you include the 95% CI for your Cohen’s d index. There is a nice website: estimationstats.com which provides the R script to calculate Cohen’s d along with bootstrap 95% CI. Here is the citation they provide: Moving beyond P values: Everyday data analysis with estimation plots. Joses Ho, Tayfun Tumkaya, Sameer Aryal, Hyungwon Choi, Adam Claridge-Chang. Nature Methods 2019, 1548-7105.

Line 326 – “… additionally tested early morphological traits apart from the growth curve…” Can you define early? Is this during the embryonic period or neonatal period?

Line 358 – 362 – Did you remove the undeveloped (infertile) eggs from the denominator?

Line 363 – It might be useful for you to provide some data as to when the embryos died. Was it early on in development or just prior to hatching?

Line 368 – add period at the end of the sentence.

Line 373 – I would change early to neonate or hatchling. When I read early, I was thinking of during the embryonic stages.

Line 374-379 – might be nice to include the 95% CI for Cohen’s d. Will give us an idea of the range of d and how “strong” the moderate effect size is.

Line 380-381 – were there differences in the sex ratios for the three treatments? Just curious.

Figure 4 figure legend – what do the boxes represent? Please provide description in the legend.

Line 438 – Do they know at what point in the embryonic development that the embryo starts to produce T3 and deiodinase? How far along into embryonic development? Can you add the timing to this sentence?

Line 446-448 – I am not sure what you mean by improving the age of embryo mortality. If an embryo dies, does changing the timing of this improve it? Because you do not provide the data for timing of embryo mortality, it is hard to judge this statement.

Figure S1 – what do the line and boxes represent? Provide in the figure legend.

Figures S5-S8 are not references in the text.

Reviewer 2 ·

Basic reporting

This study sets out to investigate the effects of maternal thyroid hormones on hatching success and longer-term metrics in Japanese quails. While I disagree that this avenue of research is particularly novel, it would nonetheless be a welcome addition to the literature. Perhaps an enhanced focus on T3 vs. T4 would help set it apart — to this end, however, there is very little in terms of their endocrinological differences discussed at the outset. To this end, I think the introduction could use a restructure to showcase i) why we care about maternal thyroid hormones in Japanese quails, or why Japanese quails (or birds in general) represent a good model to investigate these questions. While the methods were selected for ecological relevance, the overall manuscript really lacked the endocrinological and ecological context to make it matter.

Introduction. We veer into avian territory petty quickly with not setup into why looking at maternal thyroid hormones in this system would be particularly interesting. In other words, there is very little context for how an avian system can help us understand this phenomenon. I assumed its largely because embryonic THs can be more easily manipulated? Other reasons? I would recommend strengthening this link, where you go from humans to birds.

43-44. Maternal effects of thyroid hormones is not a novel concept. Of the many examples in the literatures, Wilson and McNabb (1997) show that maternal thyroid hormones influence embryonic development in the same species of quail used here — perhaps this would be relevant to in the ms.

67-79. Now we’re back to all vertebrates. I would recommend restructuring the introduction so that there is a logical flow.

104-133. There are a lot of references to studies showing answers to these questions in other species. In this context, I think the introduction would benefit from justifying why these same questions are important in the Japanese quail — is the novel angle specifically looking at differences between T3 and T4? If this is the case, I recommend tailoring the introduction to this question and contextualising differences in T3/T4 action.

Experimental design

I also echo concerns over statistical power of previous reviewers. I do not necessarily have an issue with the smaller sample sizes themselves, but rather the unequal distribution in sample size across treatment. This means the chances of finding significant differences between treatment X and control for a given metric are substantially higher for some treatments than others — i.e. small samples more prone to type II error. Further, it is difficult to appreciate the findings on hatching success without the actual numbers of unfertilized eggs for each treatment. Could this be added to the results? Did I miss it? As mentioned by a previous reviewer, I also think further replication with a strict control treatment (in addition to the sham control used here) would be useful.

Methods + Results

146-147. Just use the range — 6.6 eggs is not particularly meaningful, but the range gives an idea of how sample sizes may be spread out.

I would expect methodology this invasive to include both a control and a sham control — in other words, injection into an egg alone does not in itself represent an ecologically relevant control. A strict control would allow you to test for the “normal” hatching success rate in these birds and investigate potential interactions between injection and thyroid hormone actions.

181-183. Good. This is an important point.

183-184. This is not a problem so much as the idea that manipulated hormone levels may allow some embryos to deal with injection trauma better than others. Again, this is where a strict control treatment would have been useful. Were all eggs injected by the same individual?

209. Was this observer blind?

220. Is this the sample size of males per treatment? Probably not, right? Unless I’m mistaken, this seems misleading. It would be more appropriate to denote the sample size of males per treatment (i.e. N=6-7)?

223-224. Use the sample size notation for the above.

281. Why only 1,000 simulations? 10,000 is commonly the “unofficial” minimum and would be much more believable, especially when p hovers around 0.05.

292. I assume this takes unfertilized eggs into account?

296-299. Good.

337. I don’t understand this sentence — was the point to avoid over-parameterizing the model? Could you not have looked at each one separately?

360-362. I’m trying to understand how 21 out of 40 eggs for T3/T4 was significant, but 20 out of 40 for T4 was p>0.09? This is where more information in lines 179-181 would be more useful.

Validity of the findings

I would consider changing the title further to reflect that small sample sizes may have promoted type II errors — i.e. you did not find evidence that there are clear effects of thyroid hormones into adulthood, not that there “are no clear effects”.

External reviews were received for this submission. These reviews were used by the Editor when they made their decision, and can be downloaded below.

---

## Round 0.2 · Major Revisions

Reviewer 2 still has major concerns about the data analysis that need to be better addressed. In particular a more detailed justification for the statistical models used and the influence of unequal sample sizes. It is important that the authors effectively respond to these concerns.

Reviewer 1 ·

Basic reporting

The authors have addressed all my major concerns.

Experimental design

The authors have addressed all my major concerns.

Validity of the findings

The authors have addressed all my major concerns.

Additional comments

When Figures 2, 3, S4, and S5 are printed on a black and white printer, one can not tell which treatment each line is for.

Figure legend for S1: it references itself (Fig. S1) for the description of treatments. I think, since you added a new S1, the material to be referenced needs to move from S2 up to S1.

Reviewer 2 ·

Basic reporting

The intro is much stronger and provides great justification for the study. It now runs a little long — now that you have the general bird justification, I recommend moving the Japanese quail justification (118-127) to the methods. I also recommend condensing the text in the intro thereafter so that it focuses on hypotheses but leaves much of the more specific details for the methods.

Experimental design

I have no major issues with the design itself though I think a true control would have been very informative.

Validity of the findings

In terms of the statistical problems, I maintain there is an issue with the unequal sample sizes. The imbalance is not just between the control and treatment groups as the authors assert but also between treatment groups themselves. In the cloacal gland recession analyses, for instance, there are 2-fold and 3-fold differences in sample size between T3 vs T4 and T3 vs T3/T4 treatments, respectively. This trend is similarly true for the other analyses. These differences, coupled with the especially low sample size of the control group, means that you have more power to find statistical differences between the T3/T4 treatment and control (for instance) than for the T3 treatment and control. The same is true for T3 vs T4 to a lesser but nonetheless worrisome extent. The added issue arises from the authors reporting largely negative results (as indicated by their previous title). In short, I simply have trouble accepting that the negative results may reflect anything but a lack of statistical power (or unequal power in the experimental design).

I also now read the statistical methods more skeptically. This is a lot of effort and work-around analyses for an experimental design that should be statistically straightforward to analyse via GLM ANOVA. It does not seem to be a particularly parsimonious way of analyzing the results. I also hesitate to even use descriptive statistics when N<5.

External reviews were received for this submission. These reviews were used by the Editor when they made their decision, and can be downloaded below.

---

## Round 0.3 · accepted · Accept

Your revisions have addressed most of the reviewers' concerns. Although the questions of unequal sample sizes persist, revisions to the manuscript make it more clear for the reader the limitations of the dataset.

External reviews were received for this submission. These reviews were used by the Editor when they made their decision, and can be downloaded below.